# Quantifying Information Flow in Diffusion Models: Entropy-Guided Noise Scheduling and Mutual Information Evaluation

## Abstract

Denoising diffusion probabilistic models (DDPMs) and their variants have achieved strong performance across a wide range of tasks, from image restoration to text-to-image generation. Despite these successes, the interplay between timesteps and noise schedules in the diffusion process remains poorly understood. In particular, it is unclear how these factors shape information flow and influence the quality of the final output. This paper investigates diffusion models through the lens of information theory. We introduce an entropy-guided noise scheduling strategy and a mutual-information-based evaluation framework. First, leveraging the differential entropy of Gaussian distributions, we develop a method to compute entropy values of noisy images that are consistent with the diffusion process. Building on this, we design an entropy-guided scheduling strategy to explicitly link timesteps with noise levels during the forward process. Finally, we propose a mutual-information-based evaluation metric to assess the image restoration ability of DDPMs. Experiments on MNIST and Fashion-MNIST demonstrate the feasibility of quantifying and guiding information flow in diffusion models.

## 1 Introduction

Diffusion models have become a cornerstone of modern generative modeling, achieving state-of-the-art results in image generation, restoration, and video synthesis (Croitoru et al., 2023). These models corrupt data through a forward noising process and then learn to reverse this process to recover clean samples. Building on the pioneering work of (Sohl-Dickstein et al., 2015), (Ho et al., 2020) formalized denoising diffusion probabilistic models (DDPMs), which generate high-quality samples without relying on adversarial training.

Since then, a series of extensions have further improved efficiency and fidelity. The denoising diffusion implicit model (DDIM) (Song et al., 2020) introduced a non-Markovian forward process, enabling accelerated "jump-step" sampling. Improved DDPMs (Nichol & Dhariwal, 2021) incorporated learned variance schedules, reducing sampling complexity while enhancing likelihood performance. Score-based generative models (Song & Ermon, 2019) refined both training and sampling through optimized noise scaling strategies and adjusted Langevin dynamics. Together, these advances have established diffusion models as competitive tools for high-resolution image synthesis with increasingly efficient sampling Dhariwal & Nichol (2021); Bulat et al. (2018).

Despite this progress, a fundamental question remains unresolved:

*How do timesteps and noise schedules interact, and in what ways do they shape the diffusion process and its outcomes?*

Answering this question is crucial for principled noise schedule design and for rigorous evaluation of diffusion model performance Chen (2023); Liu & Yuan (2024).

Meanwhile, concepts from information theory Shannon (1948); Cover & Thomas (2006) have become deeply integrated into generative modeling, offering new ways to analyze and guide learning. Representative examples include InfoGAN (Chen et al., 2016), which maximizes mutual information to learn interpretable latent variables, and InfoVAE (Zhao et al., 2017), which leverages mutual

Table 1: Noise schedules used in diffusion models.

| Noise Schedule | Form of Beta ($\beta_t$) |
|---|---|
| Linear | $\beta_t = \beta_1 + \frac{\beta_T - \beta_1}{T} t$ |
| Cosine | $\beta_t = 1 - \frac{\theta_t}{\theta_{t-1}}, \theta_t = \cos^2\left(\frac{t+s}{T+s} \cdot \frac{\pi}{2}\right)$ |
| Quadratic | $\beta_t = \left(\sqrt{\beta_1} + \frac{\sqrt{\beta_T} - \sqrt{\beta_1}}{T} t\right)^2$ |
| Sigmoind | $\beta_t = \frac{1}{1+e^{-b_t}}(\beta_T - \beta_1) + \beta_1,\ b_t = 6(\frac{12}{T} - 1)$ |

information to improve posterior approximation. Furthermore, mutual information can be used in the feature selection process Hoque et al. (2016); Sulaiman & Labadin (2015). More broadly, entropy-based principles have been applied to conditional denoising, compression, and representation learning, underscoring the versatility of information-theoretic tools in deep generative models (Principe, 2010; Yang & Mandt, 2023; Zheng et al., 2022; Tishby & Zaslavsky, 2015; Shwartz-Ziv & Tishby, 2017; Kumar et al., 2025).

Motivated by these insights, this paper investigates diffusion models through an information-theoretic lens. We propose an entropy-guided noise scheduling strategy that explicitly links timesteps to noise levels, and we introduce a mutual-information-based evaluation framework to assess the restoration capacity of DDPMs. Experiments on MNIST and Fashion-MNIST demonstrate the effectiveness of our approach, showing that diffusion processes can be systematically analyzed and improved by modeling their information flow.

## 2 BACKGROUND

In this section, we briefly review diffusion models including the forward process and the related noise schedules. Diffusion models are latent variable models involving two main stages: a forward (noising) process and a reverse (denoising) process. In this study, we choose a foundational class (i.e., DDPMs) within this family as a representative and briefly describe its forward diffusion process from Ho et al. (2020). Given a data distribution $\mathbf{x}_0 \sim q(\mathbf{x}_0)$, the forward diffusion process of DDPMs, which gradually adds Gaussian noise to the data through a variance schedule $\beta_1, \beta_2, \ldots, \beta_T \in (0, 1)$, can be formulated as

$$q(\mathbf{x}_1, \mathbf{x}_2, \ldots, \mathbf{x}_T | \mathbf{x}_0) = \prod_{t=1}^{T} q(\mathbf{x}_t | \mathbf{x}_{t-1}), \quad q(\mathbf{x}_t | \mathbf{x}_{t-1}) = \mathcal{N}(\mathbf{x}_t; \sqrt{1 - \beta_t} \mathbf{x}_{t-1}, \beta_t \mathbf{I}). \quad (1)$$

As introduced in Ho XXX, the diffusion process admits sampling $\mathbf{x}_t$ at an arbitrary timestep $t$ directly depented on the input $\mathbf{x}_0$, characterized as the following closed form

$$q(\mathbf{x}_t | \mathbf{x}_0) = \mathcal{N}(\mathbf{x}_t; \sqrt{\bar{\alpha}_t} \mathbf{x}_0, (1 - \bar{\alpha}_t) \mathbf{I}), \quad \mathbf{x}_t = \sqrt{\bar{\alpha}_t} \mathbf{x}_0 + \sqrt{1 - \bar{\alpha}_t} \epsilon, \quad (2)$$

where $\alpha_t := 1 - \beta_t, \bar{\alpha}_t := \prod_{s=1}^{t} \alpha_s$. From the above analysis, the noisy data at arbitrary timestep can be measured by entropy in Information Theory.

The noise schedule has a significant impact on the training process of diffusion models, as it affects both the distribution of the noisy training set and the weights of the objective function at each noise level Chen (2023); Liu & Yuan (2024). The involved noise schedules includes Linear, Cosine, Quadratic Nichol & Dhariwal (2021) and Sigmoid [1], listed in Table 1. Here, by setting the hyperparameters $\beta_1 = 0.0001, \beta_T = 0.02, s = 0.08$, the variations of $\beta_1, \beta_2, \ldots, \beta_T$ at $T = 300$ can be observed in Figure 1.

## 3 ENTROPY-GUIDED NOISE SCHEDULING

We study in this section the relationship between timestep and noise schedule in diffusion process by introducing an entropy-guided noise scheduling. We first calculate entropy values of noisy images

---

[1] https://huggingface.co/blog/annotated-diffusion.

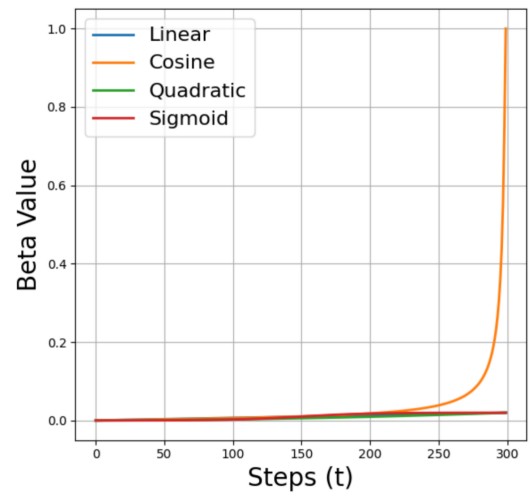

Figure 1: $\beta_t$ by different noise schedules at $T = 300$

suitable for diffusion models by the differential entropy theorem of Gaussian distribution. We then propose the entropy-guided noise scheduling to explicitly link timesteps with noise levels during the forward process.

### 3.1 COMPUTING ENTROPY VALUES OF NOISY IMAGES

We now briefly recall the theoretical foundation concerning the entropy of multivariate Gaussian distributions Cover & Thomas (2006) for computing entropy values of noisy images at each timepstep in diffusion process. The differential entropy theorem of multivariate Gaussian distributions is given as follows.

**Theorem 3.1.** *Let $X \sim \mathcal{N}(\mu, \Sigma)$ denote a multivariate Gaussian distribution, where $\mu$ represents the mean vector and $\Sigma$ denotes the covariance matrix. Then, the differential entropy $h(X)$ is defined by the following form*

$$h(X) := \frac{1}{2} \ln \left( (2\pi e)^n |\Sigma| \right),\tag{3}$$

*where $n$ denotes the dimensionality of the random variable, and $|\Sigma|$ represents the determinant of the covariance matrix.*

For an image of size $K \times K$, the corresponding random variable has a dimensionality of $n = K^2$. It is evident that entropy of a multivariate Gaussian distribution depends solely on the dimensionality of this distribution and its covariance matrix. Based on the above theorem, the following corollary can be derived.

**Corollary 3.1.** *Let $X$ and $X_0$ be $n$-dimensional continuous random variables. If $X \sim \mathcal{N}(\sqrt{\bar{\alpha}} \bar{X}_0, (1 - \bar{\alpha})\mathbf{I})$, then the differential entropy $h(X)$ is given by*

$$H(X) := \frac{n}{2} \ln \left( (2\pi e)(1 - \bar{\alpha}) \right).\tag{4}$$

According to equation (2) , the random variable $\mathbf{x}_t$ at timestep $t$ in the diffusion process is modeled as a multivariat Gaussian distribution with mean $\sqrt{\bar{\alpha}_t}\mathbf{x}_0$ and covariance matrix $(1 - \bar{\alpha}_t)\mathbf{I}$. In fact, the random variable $\mathbf{x}_t$ is typically discrete on practical applications. Consequently, wiht equations (4 , we can calculate entropy value of $\mathbf{x}_t$ by the following form

$$H(\mathbf{x}_t) \approx \frac{n}{2} \ln \left( (2\pi e)(1 - \bar{\alpha}_t) \right).\tag{5}$$

From the perspective of information entropy, it becomes evident that when analyzing multi-dimensional Gaussian distributions, the mean vector does not require excessive attention. The entropy can be directly derived using the covariance matrix and the dimensionality of the distribution.

Therefore, it can be concluded that during the forward process, the image undergoes a gradual addition of Gaussian noise, resulting in a progressive increase in information entropy. As $t \to \infty$, the entropy value of the final noisy image $\mathbf{x}_t$ converges to $\frac{n}{2} \ln(2\pi e)$.

## 3.2 SCHEDULING AND EXPERIMENTS

We propose in this section an entropy-guided noise scheduling strategy to explore the explicitly link timesteps with noise levels during the forward process. The resulting strategy can be described as follows

(i) Use equation (5) to calculate entropy values of noisy data yielded by different noise schedules based forward process;

(ii) Find the corresponding timestep of each schedule by the same or similar entropy value;

(iii) Obtain the combination of noise schedules according to entropy value and timestep of each schedule.

This leads to achieve the combination of different noise schedules at the corresponding timesteps, and contributes to the efficient and reliable implementation of diffusion models. To this end, we next conduct experiments to demonstrate the feasibility of the proposed strategy in terms of three cases: different noise schedules at the same timesteps, noise schedule with different timesteps, and entropy-guided inverse process. The used dataset is MNIST handwritten dataset with the size $28 \times 28$ image in this experiment. Entropy values of noise images by different noise schedules based forward processes on this dataset is given in Figure 2.

### 3.2.1 DIFFERENT NOISE SCHEDULES AT THE SAME TIMESTEPS $T$

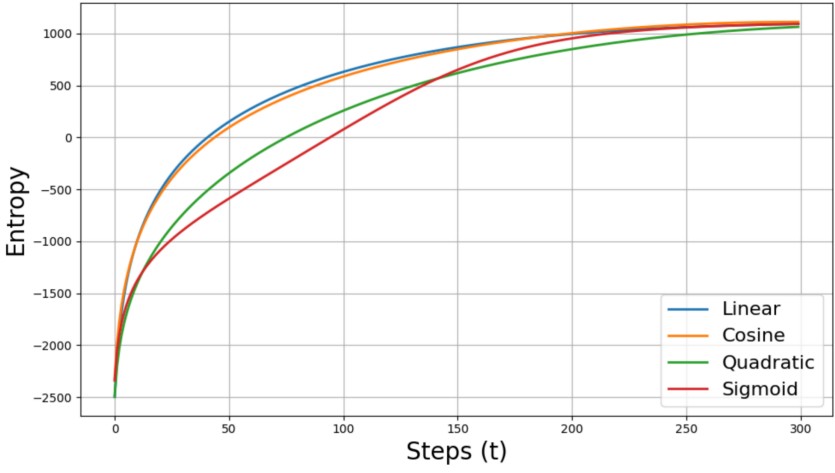

Figure 2: Entropy values for the MNIST handwritten digit dataset by different noise schedules at $T = 300$.

Initially, a correlation is established between Linear and Cosine noise schedules under conditions of similar entropy value variation. During the forward diffusion process, noise from Linear schedule is applied to generate noisy images $\mathbf{x}_{50}, \mathbf{x}_{90}, \mathbf{x}_{150}$. Based on their respective entropy values, an appropriate step size from Cosine is selected for reverse sampling, specifically corresponding to steps 55, 96, and 155. The inverse sampling results were evaluated in terms of cosine similarity (Similarity), mean squared error (MSE), Euclidean distance (ED), and structural similarity index (SSIM)Wang et al. (2004) by comparing them with the original image $\mathbf{x}_0$. The corresponding results are summarized in Table 2.

Subsequently, we conduct analogous analyses on Cosine and Sigmoid schedules, which exhibit notable differences in entropy values (Figure 2). Specifically, we select images $\mathbf{x}_{18}, \mathbf{x}_{49}, \mathbf{x}_{110}$ obtained after applying Sigmoid noise at steps 18, 49, and 110, respectively. Corresponding to entropy values

Table 2: Evaluation of sampling results, where FL50 represents 50 steps of the forward diffusion based on Linear schedule, RC55 denotes 55 steps of the reverse process with Cosine.

| Schedule | MSE | ED | SSIM | SIMILARITY |
|---|---|---|---|---|
| FL50+RC55 | 0.0073 | 0.0265 | 0.90498 | 0.9649 |
| FL90+RC96 | 0.0184 | 0.0413 | 0.82312 | 0.9121 |
| FL150+RC155 | 0.0710 | 0.1013 | 0.53299 | 0.7229 |
| FL50+RL50 | 0.0067 | 0.0279 | 0.87162 | 0.9679 |
| FL90+RL90 | 0.0183 | 0.0440 | 0.79427 | 0.9143 |
| FL150+RL150 | 0.0750 | 0.1098 | 0.49374 | 0.7175 |

Table 3: Evaluation of sampling results, where FS18 represents 18 steps of the forward diffusion using Sigmoid.

| Schedule | MSE | ED | SSIM | SIMILARITY |
|---|---|---|---|---|
| FS18+RC50 | 0.0018 | 0.0176 | 0.8957 | 0.9913 |
| FS49+RC100 | 0.0063 | 0.0250 | 0.9062 | 0.9702 |
| FS110+RC150 | 0.0242 | 0.0485 | 0.7849 | 0.8885 |
| FS18+RS18 | 0.0019 | 0.0171 | 0.9149 | 0.9910 |
| FS49+RS49 | 0.0063 | 0.0254 | 0.9044 | 0.9698 |
| FS110+RS110 | 0.0250 | 0.0503 | 0.7781 | 0.8877 |

of these noisy images, we then identify steps $50, 100$, and $150$ in the cosine for further comparison. A quantitative evaluation of the inverse sampling results is presented in Table 3.

### 3.2.2 Noise schedule with different timesteps $T$

We further investigate the potential of entropy values by conducting experiments on the same noise schedule with different timesteps $T$.

We conducted comparative experimental analyses on DDPM models employing Cosine and Sigmoid noise schedules. First, we selected images $\mathbf{x}_{40}, \mathbf{x}_{75}, \mathbf{x}_{150}$ via Sigmoid, which were generated under $T1 = 300$ with steps $40, 75$, and $150$, respectively. Subsequently, based on the entropy values observed at $T2 = 500$ and $T3 = 1000$, we determined the corresponding noise addition steps in the forward process: $48, 100, 220$ for $T2$, and $56, 137, 372$ for $T3$. The detailed sampling performance are summarized in Table 4.

Table 4: Evaluation of sampling results by Sigmoid schedule with different timesteps $T1 = 300, T2 = 500, T3 = 1000$, where FS40+RS40(T1) denotes 40 steps of the forward and reverse process under the timesteps $T1$, focusing on images generated by adding noise at a specific timesteps $T1 = 300$.

| Schedule | MSE | ED | SSIM | SIMILARITY |
|---|---|---|---|---|
| FS40+RS40 (T1) | 0.0014 | 0.0157 | 0.9187 | 0.9931 |
| FS75+RS75 (T1) | 0.0035 | 0.0205 | 0.9193 | 0.9833 |
| FS150+RS150 (T1) | 0.0254 | 0.0510 | 0.7725 | 0.8865 |
| FS48+RS48 (T2) | 0.0016 | 0.0232 | 0.7955 | 0.9920 |
| FS100+RS100 (T2) | 0.0037 | 0.0283 | 0.7870 | 0.9818 |
| FS220+RS220 (T2) | 0.0295 | 0.0600 | 0.6803 | 0.8664 |
| FS56+RS56 (T3) | 0.0014 | 0.0153 | 0.9267 | 0.9932 |
| FS137+RS137 (T3) | 0.0034 | 0.0201 | 0.9229 | 0.9835 |
| FS372+RS372 (T3) | 0.0254 | 0.0516 | 0.7657 | 0.8868 |

Additionally, we explore the application of small-step DDPMs to approximate the reverse sampling process of large-step DDPMs. Furthermore, we investigate the feasibility of substituting these models with DDPMs employing distinct noise schemes across varying step sizes. We choose images $\mathbf{x}_{167}, \mathbf{x}_{335}, \mathbf{x}_{435}$ by Cosine schedule at $T3 = 1000$ with steps $167, 335, 435$, respectively. Based on the entropy values observed at $T1 = 300$ and $T2 = 500$, we determine the corresponding steps in the forward process: $50, 100, 130$ for $T1$, and $85, 170, 218$ for $T2$. Furthermore, we identify steps $76, 145, 178$ in the forward process of Quadratic schedule. Inverse sampling was then performed using these selected steps. Table 5 shows the evaluation of the above sampling results.

Table 5: Evaluation of sampling results with Cosine and Quadratic schedules, for images generated by adding noise at a specific step of $T3 = 1000$.

| Schedule | MSE | ED | SSIM | SIMILARITY |
|---|---|---|---|---|
| FC167+ RC167 (T3) | 0.0046 | 0.0219 | 0.9219 | 0.9780 |
| FC335+ RC335 (T3) | 0.0171 | 0.0409 | 0.8189 | 0.9192 |
| FC435+ RC435 (T3) | 0.0603 | 0.0915 | 0.5752 | 0.7610 |
| FC50+ RC50 (T1) | 0.0049 | 0.0403 | 0.7049 | 0.9753 |
| FC100+ RC100 (T1) | 0.0169 | 0.0699 | 0.6018 | 0.9147 |
| FC130+ RC130 (T1) | 0.0389 | 0.0939 | 0.4782 | 0.8051 |
| FC85+ RC85 (T2) | 0.0047 | 0.0431 | 0.7035 | 0.9768 |
| FC170+ RC170 (T2) | 0.0158 | 0.0595 | 0.6290 | 0.9209 |
| FC218+ RC218 (T2) | 0.0438 | 0.0870 | 0.4983 | 0.7838 |
| FQ76+ RQ76 (T1) | 0.0049 | 0.0245 | 0.8895 | 0.9760 |
| FQ145+ RQ145 (T1) | 0.0190 | 0.0443 | 0.7982 | 0.9137 |
| FQ178+ RQ178 (T1) | 0.0564 | 0.0885 | 0.5803 | 0.7676 |

### 3.2.3 ENTROPY-GUIDED INVERSE PROCESS

In the previous experiments, it has been demonstrated that images corrupted with arbitrary noise schedules or timesteps can be reconstructed through entropy value based selection to identify the corresponding timestep in the noise schedule. This approach is independent of the specific noise schedule or timestep configuration, and achieves performance comparable to that of the original DDPM model with its native noise schedule and timestep.

We now investigate whether the entire reverse diffusion process can be effectively guided by entropy. In particular, we examine whether a DDPM framework, employing entropy-driven selection of hybrid noise schedules or variable noise step sizes, is capable of reconstructing handwritten digit images from random Gaussian noise through the reverse sampling.

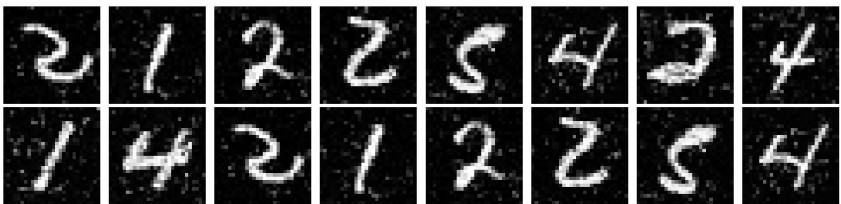

Figure 3: Sampling results at $T = 300$ by the entropy-guide schedule of Quadratic followed by Cosine.

Based on the calculated entropy values, it can be observed that for $T = 300$, the entropy value of the noisy image in Cosine at step $t = 130$ is comparable to that of the noisy image in the Quadratic at step $t = 178$. Therefore, a noise sample $\mathbf{x}_0^*$ drawn from a standard Gaussian distribution is utilized for the reverse sampling process. The original DDPM models employ Cosine and Quadratic noise schemes individually. Specifically, the procedure first employs Quadratic schedule to perform inverse sampling for 122 steps starting from step 300, resulting in $\mathbf{x}_{\text{qua178}}$. Subsequently, Cosine

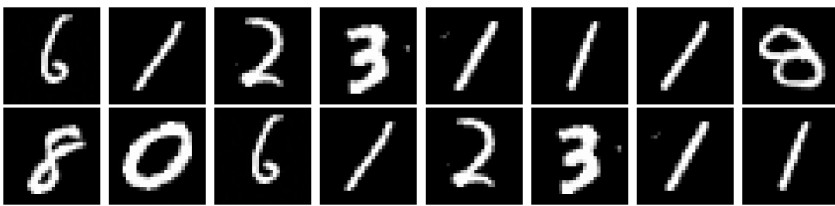

Figure 4: Sampling results at $T = 300$ using the entropy-guide schedule of Cosine followed by Quadratic.

is applied to further conduct inverse sampling from step 130 to step 0, thereby generating the final image (comprising a total of 252 sampling steps). The experimental results are presented in Figure 3. Through Tables 7-6 and Figure 4, it can be observed that although the entropy-guided mixed noise schedule in DDPM inverse sampling reduces the sampling time required to generate handwritten digits, the generation quality is marginally inferior to that of the original DDPM models employing Cosine and Quadratic noise schemes individually.

Table 6: The Inception Score (IS) and Fréchet Inception Distance (FID) of the DDPM-generated results under different noise schedules at three timesteps.

| Metrics | Schedule | T=300 | T=500 | T=1000 |
|---|---|---|---|---|
| IS | Linear | 1.000816 | 1.000850 | 1.001001 |
| | Cosine | 1.000957 | 1.001039 | 1.000876 |
| | Quadratic | 1.000860 | 1.000873 | 1.000854 |
| | Sigmoid | 1.000921 | 1.001058 | 1.000865 |
| FID | Linear | 126.730 | 550.910 | 210.436 |
| | Cosine | 313.253 | 550.054 | 156.361 |
| | Quadratic | 138.863 | 134.126 | 155.218 |
| | Sigmoid | 140.521 | 249.828 | 118.942 |

Table 7: Evaluation of sampling resluts by two entropy-guided reverse process at $T = 300$, including Quadratic-Cosine and Cosine-Quadratic.

| Metrics | Schedule | T=300 |
|---|---|---|
| IS | Quadratic-Cosine | 1.000604 |
| | Cosine-Quadratic | 1.000921 |
| FID | Quadratic-Cosine | 1002.769 |
| | Cosine-Quadratic | 118.455 |

However, it is noteworthy that when the operational order is altered—specifically, by first sampling $\mathbf{x}_0^*$ from a standard Gaussian distribution, followed by 170 reverse steps using Cosine schedule starting from $T = 300$ to generate $\mathbf{x}_{130}^{cos}$, and subsequently applying Quadratic to reverse sample from $T = 178$ to 0 to generate the final image (totaling 348 sampling steps)—the resulting outputs, as illustrated in Figure 4. IS Reed et al. (2016) and FID Salimans et al. (2016) are listed in Table 7. These results demonstrate a performance level comparable to or slightly improved over those obtained using individual noise schemes as summarized in Table 6.

## 4 MUTUAL INFORMATION EVALUATION

In this section, we develop a mutual information based evaluation method. To calculate the value of mutual infomation, the probability density of image pixel values is estimated through the binning histogram method, which allows for the derivation of the entropy terms $H(X), H(Y), H(X, Y)$.

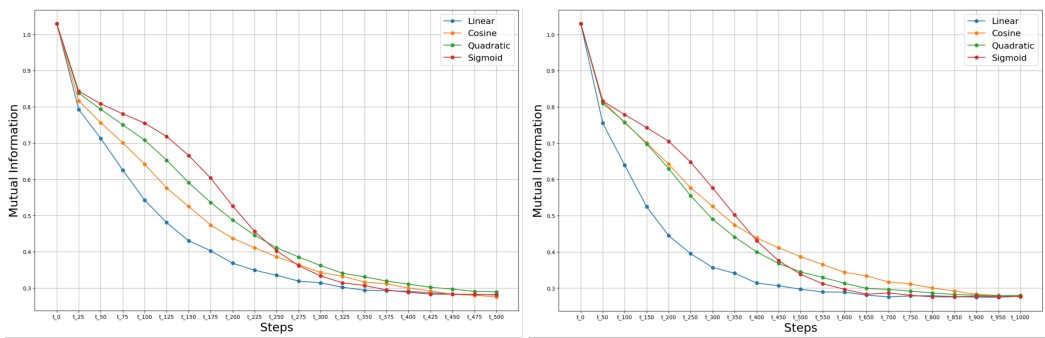

Figure 5: Mutual information during the diffffferent noise schedule based forward process at $T = 500$ (left) and $T = 1000$ (right) for MNIST dataset.

The mutual information $I(X;Y)$ can be computed by the following algorithm in A.2. With the algorithm, the mutual information between the MNIST dataset and itself is measured as 1.030006. The level of image complexity varies across different datasets, resulting in differing information contents. For instance, the Fashion MNIST dataset, which contains more complex images compared to MNIST, exhibits the mutual information value of 2.410689—significantly higher than that n of MNIST.

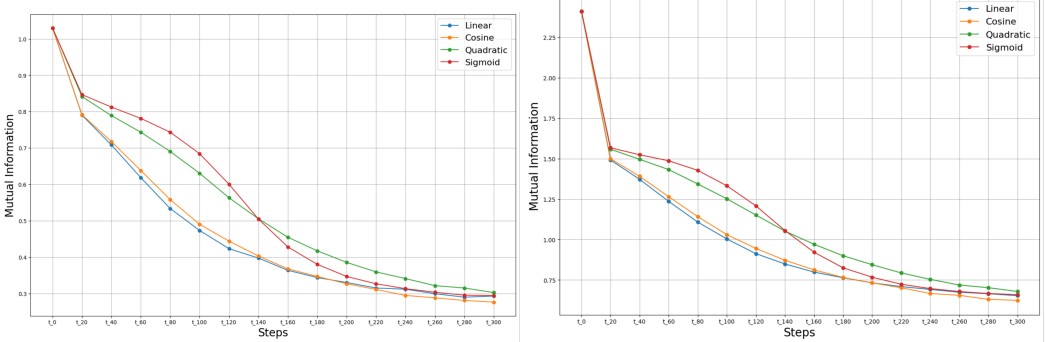

Figure 6: Mutual information during the different noise schedules based forward process at $T = 300$ for MNIST (left) and Fashion MNIST datasets (right).

We first utilize various variance scheduling strategies and train the DDPM models on the MNIST dataset. During the forward diffusion process, various variance scheduling strategies introduce noise into the MNIST, progressively degrading the image quality. We then employ mutual information as a quantitative criterion to systematically compare the effectiveness of Similarity, MSE, ED, and SSIM in evaluating image restoration performance throughout the forward process for two datasets. As illustrated in Figures 7, 8, 9, and 10, the MNIST handwritten dataset and the Fashion MNIST dataset are presented alongside the variations of their respective evaluation metrics throughout the forward diffusion process.

Compared with the above metrics, mutual information inherently incorporates the entropy of dataset within its computation method. Mutual information inherently depends on the complexity of the underlying dataset, leading to varying maximum mutual information values. Consequently, under identical noise schemes, the image restoration performance of DDPMs can exhibit significant differences across datasets. Notably, mutual information effectively captures the inherent complexity of images across diverse datasets through its mathematical formulation, thereby offering more effective guidance for image restoration tasks.

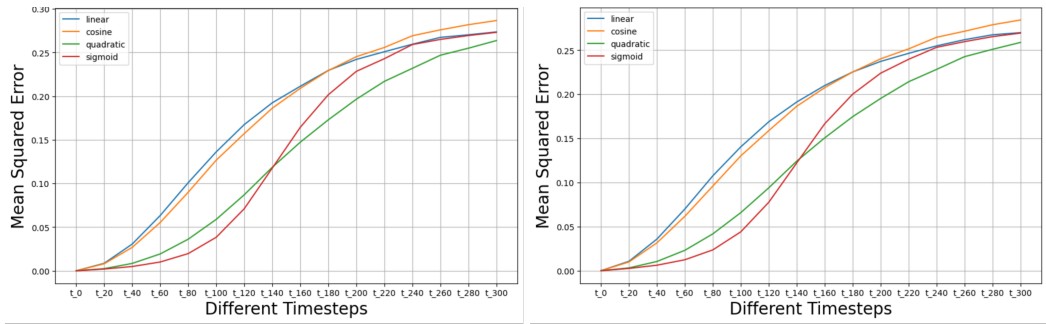

Figure 7: MSE during the forward process under different noise schedules on the MNIST (left) and Fashion MNIST datasets (right).

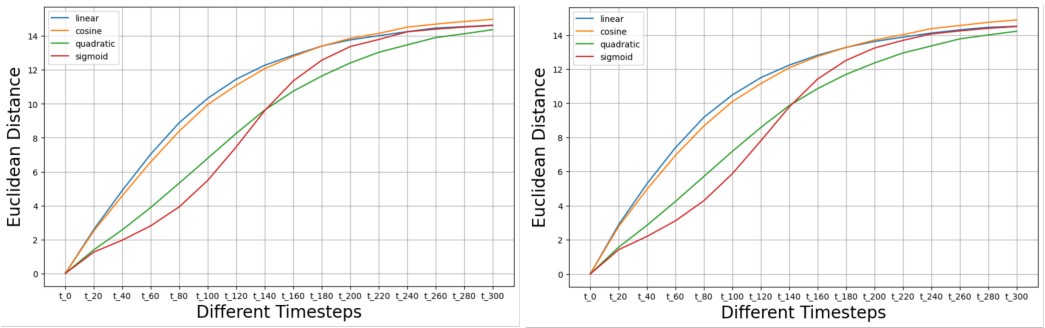

Figure 8: ED during the forward process under different noise schedules on the MNIST (left) and Fashion MNIST datasets (right).

## 5 CONCLUSION

This work investigated the information flow in diffusion models by introducing an entropy-guided noise scheduling strategy and developing a mutual-information-based evaluation method. Experiments were conducted to examine the relationship between noise schedules and timesteps, and the results indicated that combining different noise schedules at corresponding timesteps was both feasible and effective. In evaluating the restoration capacity of DDPMs, the proposed mutual-information-based metric demonstrated greater sensitivity than conventional measures, particularly in capturing complexity variations across different datasets.

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

## A APPENDIX

### A.1 EXPERIMENTAL SETUP AND PARAMETER CONFIGURATION

This experiment was carried out on a system featuring dual NVIDIA RTX 4090 GPUs, 128GB of RAM, and an Intel i7-13900K processor. The DDPM implementation was based on the deep learning code repositories for research papers LabML.ai (2024) provided by LabML.ai on GitHub. Furthermore, the model underwent fine-tuning based on the DDPM course implementation provided by Hugging Face Hugging Face (2025).The implementation of information entropy and mutual information in the experiment was developed in accordance with the respective procedural steps.

The experimental data employed in this study is the MNIST handwritten digit dataset, which consists of $70,000$ grayscale images representing digits from 0 to 9. Each image has a resolution of 28x28 pixels and is encoded using grayscale intensity values within the normalized range [0, 1], where 0 corresponds to black, 1 corresponds to white, and intermediate values denote varying shades of gray.

The DDPM model in this experiment employs the variance schedules described in the previous chapter, including linear noise, cosine noise, square root linear interpolation noise, and Sigmoid variance. The hyperparameters remain consistent with the specified values: $\beta_0 = 0.0001, \beta_T = 0.02, s = 0.08$. The DDPM models trained under different noise schemes maintain structural consistency. The total time steps (T) are configured based on specific experimental objectives. For grayscale image processing, all models are trained for a fixed duration of 1000 epochs, with a standardized batch size of 128.

### A.2 MUTUAL INFORMATION ALGORITHM AND EXPERIMENTS

---

**Algorithm 1** Compute the Mutual Information

---

1: **Input:** Dataset $X = \{x_1, x_2, \ldots, x_N\}$ and $Y = \{y_1, y_2, \ldots, y_N\}$
2: **Output:** Mutual information $I(X;Y)$
3: Partition $X$ into $m$ intervals, obtaining the count $C_j$ for each bin $(j = 1, 2, \ldots, m)$
4: Compute the marginal probability distribution:

$$p(x_j) = \frac{C_j}{N};$$

5: Similarly, partition $Y$ into $m$ intervals and compute the marginal probability distribution:

$$p(y_j) = \frac{C_j}{N};$$

6: Construct the joint histogram by counting $C_{ij}$ for each pair $(x_i, y_j)$, where $i, j = 1, 2, \ldots, m$. The joint probability distribution is:

$$p(x_i, y_j) = \frac{C_{ij}}{N};$$

7: Compute the entropies:

$$H(X) = -\sum_{i=1}^{m} p(x_i) \log p(x_i), \quad H(X, Y) = -\sum_{i=1}^{m} \sum_{j=1}^{m} p(x_i, y_j) \log p(x_i, y_j);$$

8: Calculate the mutual information:

$$I(X;Y) = H(X) + H(Y) - H(X, Y).$$

---

## B REPRODUCIBILITY STATEMENT

We provide all necessary details to support reproducibility. All experiments are conducted on publicly available datasets, and the model architectures, hyperparameters, training protocols, and evalu-

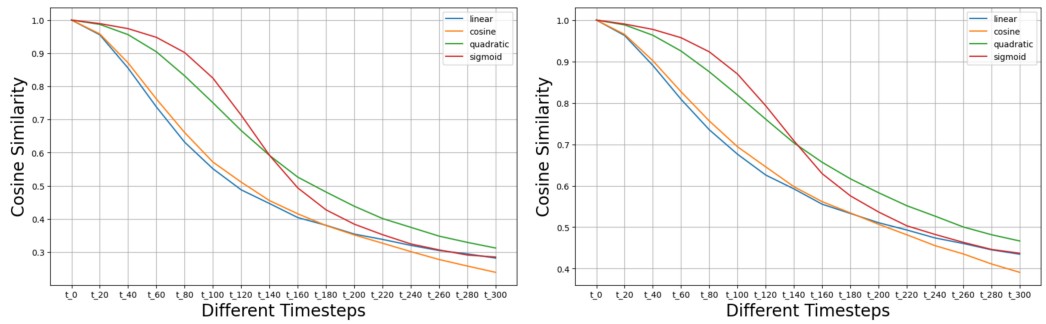

Figure 9: Cosine similarity during the forward process under different noise schedules on the MNIST (left) and Fashion MNIST datasets (right).

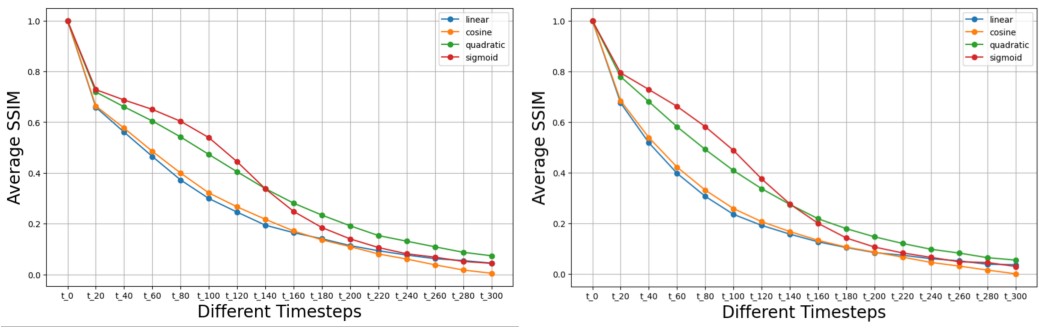

Figure 10: SSIM during the forward process under different noise schedules on the MNIST (left) and Fashion MNIST datasets (right).

ation metrics are specified in the paper. We will release our codebase, training scripts, and pretrained checkpoints on GitHub upon acceptance.

## C    THE USE OF LARGE LANGUAGE MODELS (LLMS)

We used large language models only for light editorial assistance during manuscript preparation (grammar and wording refinement, minor style/formatting suggestions). No LLMs were used for research ideation, dataset curation, modeling, experiment design, analysis, or drafting substantive sections.

