# OpenReview forum: "Quantifying Information Flow in Diffusion Models: Entropy-Guided Noise Scheduling and Mutual Information Evaluation"
_ICLR.cc/2026/Conference — ICLR 2026 Conference Withdrawn Submission_

### Official Review · Reviewer_wH5s · 2025-10-19

**Soundness:** 3
**Presentation:** 2
**Contribution:** 2
**Rating:** 4
**Confidence:** 4

**Summary:**

This manuscript attempts to analyze diffusion models from an information-theoretic perspective and presents two main contributions: first, it proposes an *entropy-guided noise scheduling* strategy, derived from the differential entropy of Gaussian distributions, as a unified measure of noise levels. This strategy allows for the alignment or mixing of different noise schedules during reverse sampling. Second, it introduces a *mutual information-based evaluation framework* aimed at using mutual information to assess the image restoration capacity of models. The authors validate the feasibility of these methods through experiments on the MNIST and Fashion-MNIST datasets.

**Strengths:**

1.  The advantage of this paper lies in the use of differential entropy $H(x_t)$ as a unified *scale* to quantify noise levels. As shown in the formula, entropy is a monotonic function of $\bar{\alpha}_t$, providing a clear theoretical basis for comparing and aligning different noise schedules (e.g., Linear vs. Cosine) or varying total timesteps (e.g., T=300 vs. T=1000).

2. The paper demonstrates the possibility of *mixing* different noise schedules during a single reverse sampling process. For instance,  starting from a standard Gaussian noise, the image is first denoised using a Quadratic schedule, and then the sampling process switches to a Cosine schedule at an equivalent entropy point, still producing recognizable images. This experiment supports the idea that the state of the diffusion process (defined by entropy or $\bar{\alpha}_t$) is more important than the specific noising path taken.

**Weaknesses:**

1. The paper merely uses this framework to align existing schedules, rather than applying information theory principles to design a new, more effective scheduling strategy.


2. The experimental validation lacks convincing evidence, as the baseline model performance is relatively poor. Table 6 shows that the FID score of DDPM on MNIST ranges from 118.9 to 550.9, which appears unusually high.

3. While the paper claims that the metrics capture 'image complexity,' no evidence is provided to support that these findings apply to more complex, higher-resolution, color datasets such as CIFAR-10, CelebA, or ImageNet.

**Questions:**

This work provides a valuable investigation into using entropy and information theory to guide and evaluate diffusion models. However, the discussion appears to be missing a comparison to a highly relevant and recent body of work, [1], which is the conference version of the initial submission for the ICLR 2025 paper. It would be helpful if the authors could clarify how their work differs from the following methods:

[1] Li, S., et al., EVODiff: Entropy-aware Variance Optimized Diffusion Inference, NeurIPS 2025. (An earlier version titled Improving Denoising Diffusion with Efficient Conditional Entropy Reduction was submitted to ICLR 2025 in 2024).

---

### Official Review · Reviewer_JSNp · 2025-10-25

**Soundness:** 2
**Presentation:** 1
**Contribution:** 1
**Rating:** 2
**Confidence:** 5

**Summary:**

This paper proposes an information-theoretic framework for analyzing diffusion models, introducing entropy-guided noise scheduling and mutual-information-based evaluation. The authors derive analytical expressions for computing entropy values of noisy images during the diffusion process and use these to develop a scheduling strategy that links timesteps with noise levels across different schedules. They also propose mutual information as an evaluation metric for assessing image restoration capacity. Experiments on MNIST and Fashion-MNIST demonstrate the feasibility of quantifying information flow and combining different noise schedules based on entropy matching.

**Strengths:**

[1] The application of entropy and mutual information to analyze diffusion processes provides a fresh viewpoint on noise scheduling and evaluation.

[2] The entropy-matching approach for combining different noise schedules offers a principled method for hybrid schedule design (Sec 3.2.1-3.2.3).

[3] The paper systematically compares multiple noise schedules (Linear, Cosine, Quadratic, Sigmoid) across different timesteps and provides extensive quantitative results (Tables 2-7).

[4] The derivation of entropy expressions for noisy images (Theorem 3.1, Corollary 3.1) is mathematically sound and well-explained.

**Weaknesses:**

[1] Experiments are confined to MNIST and Fashion-MNIST (28×28 grayscale), which are relatively simple compared to modern diffusion model applications. No results on larger or more complex datasets like CIFAR or ImageNet.

[2] The proposed methods show only small improvements over standard approaches, with some hybrid schedules actually performing worse (Table 7 FID scores).

[3] The entropy computation is straightforward for Gaussian distributions, and the mutual information implementation uses basic histogram methods without addressing the challenges of continuous-valued images.

[4] The evaluation focuses on low-level metrics but lacks analysis of how the proposed methods affect final generation quality or sampling efficiency in practical scenarios.

**Questions:**

[1] How would the entropy-guided scheduling approach scale to higher-resolution color images and more complex datasets?

[2] Could you provide more insight into why some hybrid schedules (like Quadratic-Cosine) perform significantly worse than individual schedules?

[3] Have you considered more sophisticated mutual information estimators (like kernel-based methods) that might work better for continuous image data?

[4] What are the computational overheads of the proposed methods, and how do they affect training and sampling times compared to standard approaches?

---

### Official Review · Reviewer_EXCn · 2025-11-01

**Soundness:** 1
**Presentation:** 2
**Contribution:** 1
**Rating:** 2
**Confidence:** 5

**Summary:**

This paper studies diffusion models from an information-theoretic perspective by introducing entropy-guided noise scheduling and mutual-information-based evaluation. The authors argue that entropy can quantify information flow during the diffusion process and propose using it to align or combine different noise schedules. Experiments on MNIST and Fashion-MNIST support the feasibility of this analysis.

**Strengths:**

* The paper is clearly written and follows a well-organized experimental procedure.
* The attempt to interpret diffusion models via entropy and mutual information is conceptually interesting and could inspire further analytical work.
* The idea of unifying different noise schedules under a common metric may have some heuristic value for schedule design.

**Weaknesses:**

* The key observation—($H(x_t) = \frac{n}{2}\ln((2\pi e)(1-\bar{\alpha}_t))$)—is trivial, as it merely reflects the log-scale of predefined Gaussian noise variance. It does not yield any new insight into diffusion dynamics.
* The entropy differences across noise schedules are expected, since they directly follow from how the schedules are defined.
* The proposed “entropy matching” and hybrid scheduling are empirically motivated rather than theoretically derived, and their benefit may not generalize beyond toy datasets.
* Experiments are restricted to MNIST and Fashion-MNIST, limiting the relevance and transferability of the findings to realistic generative modeling setups.
* Overall, the work reiterates well-known properties of diffusion processes without producing substantive analytical or methodological novelty.

**Questions:**

* Could the authors formalize the theoretical connection (if any) between entropy change and optimal schedule design, beyond empirical matching?
* Would learning noise schedules adaptively, e.g., by optimizing entropy progression or dataset-dependent information retention, yield more meaningful insights?
* Have the authors tested whether their entropy-based analysis still holds for other prediction targets (e.g., (x_0), (\epsilon), or velocity prediction) or higher-resolution datasets?

---

### Official Review · Reviewer_yse5 · 2025-11-03

**Soundness:** 1
**Presentation:** 1
**Contribution:** 1
**Rating:** 0
**Confidence:** 5

**Summary:**

This reads more like an experimental record than an article. It's simply a compilation of experimental data, lacking any explanation of the experiment's motivation, analysis of the results, or clear conclusions.

**Strengths:**

NA

**Weaknesses:**

NA

**Questions:**

NA

---

### Note · Authors · 2025-11-14

I have read and agree with the venue's withdrawal policy on behalf of myself and my co-authors.